# Gender Differences in Socio-Demographic Factors Associated with Pre-Frailty in Japanese Rural Community-Dwelling Older Adults: A Cross-Sectional Study

**DOI:** 10.3390/ijerph20021091

**Published:** 2023-01-07

**Authors:** Aki Shibata, Asuka Suzuki, Kenzo Takahashi

**Affiliations:** 1Graduate School of Public Health, Teikyo University, 2 11 1 Kaga Itabashi ku, Tokyo 173 8605, Japan; 2Department of Nursing, Saitama Prefectural University, 820 Sannomiya Koshigaya shi, Saitama 343 8540, Japan

**Keywords:** frailty, pre-frailty, social factors, gender differences

## Abstract

Purpose: To prevent frailty, it is necessary to focus on pre-frailty and consider preventive interventions that incorporate social aspects. This study aimed to explore socio-demographic associations with pre-frailty, focusing on modifiable social factors among community-dwelling older adults in a rural Japanese city. Methods: We conducted a self-administered survey on social, physical, and mental factors, and basic attributes, in September 2021. Respondents were classified as frail, pre-frail, or healthy according to their The Kihon Checklist scores. Of the 494 valid responses, 93 respondents classified as pre-frail and 110 as healthy were analyzed. The socio-demographic associations with pre-frailty were investigated by multiple logistic regression, and interactions between gender and other factors were examined. Results: The analysis identified that for both genders, no-community participation in middle age (odds ratio [OR], 1.84; 95% confidence interval [CI], 1.16–2.92) was found to be a social factor associated with pre-frailty. Having friends who listen to one’s concerns (OR, 2.54; 95% CI, 1.26–5.10) was a factor for women. Conclusions: This study showed that modifiable social factors associated with pre-frailty were community involvement and being able to share concerns with friends. The findings suggest the need for support that emphasizes social aspects to prevent pre-frailty.

## 1. Introduction

Global population aging has attracted a lot of interest and raised concerns regarding frailty in older people, which leads to reduced physical capacity and increased mortality. Japan has the highest life expectancy in the world and has developed a national policy to address frailty. As part of this policy, frailty health checkups have been conducted in Japan since April 2020. Frailty is a multifactorial syndrome characterized by a decline in muscle strength, endurance, and physiological function, and increased vulnerability to negative health outcomes such as physical dysfunction and death [1].

Although the risk of frailty increases with age, the effects of frailty can be mitigated with appropriate interventions [2]. Because frailty is a reversible process, it is important to support recovery from frailty among older people in aging societies such as Japan. In addition, frailty is characterized by a temporal continuum from healthy to pre-frailty and then to frail. Pre-frailty is defined as a condition that predisposes the individual to frailty and usually precedes frail [3]. Therefore, as well as supporting recovery from frailty, a focus on pre-frailty, the stage before frail, is important in order to initiate timely interventions to prevent the transition to frailty. We considered that focusing on pre-frailty and early interventions to prevent the transition to frailty were a key component to minimize the social influence of frailty. The prevalence of pre-frailty is high among older people living in the community, indicating the need to focus not only on frailty but also on pre-frail [4]. Furthermore, the factors that affect improvement, and thus appropriate intervention methods, can differ between frailty and pre-frailty [5], and consideration of pre-frailty-specific support to prevent the transition to frailty is needed.

Frailty is a multifaceted concept that includes mental, psychological, and social vulnerability, as well as physical vulnerability [6]; therefore, efforts to address frailty should be multidimensional. However, most reports on the effectiveness of frailty prevention interventions have focused on the physical aspects, such as exercise and nutrition [7,8]. Social factors that affect frailty include poor economic conditions [9], social networks, social support, social participation, and relationships with neighbors [5,10,11,12]. In previous studies on gender differences in frailty factors, it has been reported that anxiety increases the risk of frailty in women and functionality suppresses the risk of frailty in men [13]. They also reported low-to-moderate correlations between social vulnerability and frailty, with stronger correlations observed for women than for men [14]. Furthermore, it has been noted that there are gender differences in frailty, such as men having “few relatives and neighbors” and “not participating in helping others”, and women having “few social contacts with relatives” and “not participating in community activities,” which affect their frailty. However, not many reports have focused on gender differences in social factors related to frailty. However, there are few reports focusing on gender differences in social factors associated with frailty, although it has been pointed out that there are gender differences in such factors, such as men having “few relatives and neighbors” and “not participating in helping others”, and women having “few social contacts with relatives” and “not participating in community activities” [15]. Furthermore, there are few reports on social factors affecting pre-frailty [5], and to the best of our knowledge, no studies have focused on differences by gender.

Given frailty can be mitigated, we believe that focusing on pre-frailty prevention and conducting multifaceted research that includes social aspects could be useful in informing measures to prevent or improve the transition to frailty, taking into account gender differences. Therefore, the purpose of this study was to examine possible factors that affect pre-frailty, with a focus on social aspects, including gender differences.

## 2. Materials and Methods

### 2.1. Study Design and Participants

This was a cross-sectional study. The survey population was 900 stratified by randomly selected residents of Nagatoro town, Saitama Prefecture, Japan, aged ≥65 years. Nagatoro has a population of 6807 and an aging rate of 39.6%, compared with 28.6% for Japan as a whole [16]. In Nagatoro town, where the population is expected to age further in the future, measures to prevent frailty are an urgent issue. In a depopulated area with an aging population such as Nagatoro town, it is expected that many of the residents’ life backgrounds are similar, and the results of this study may be adapted if a certain level of evaluation is obtained in similar areas. The survey participants were mailed a self-administered questionnaire. The survey period was from August to September 2021. To examine individual attributes associated with pre-frailty, individuals judged as pre-frail or healthy according to the frailty criteria described below were included in the analysis, and those judged as frail were excluded from the analysis.

### 2.2. Survey Items

#### 2.2.1. Frailty

The Kihon Checklist (KCL) was used to assess frailty. The KCL was published by the Ministry of Health, Labour, and Welfare in Japan in 2006 [17] and has been validated to assess frailty [18]. The checklist has been translated into English [19], Portuguese [20], Spanish [21,22], Chinese [23], Thai [24], Turkish [25], and other languages.

The KCL is a comprehensive assessment of frailty that comprises 25 questions on physical (motor function, nutritional status, and oral function), mental (cognitive function, and depressed mood), and social (being housebound) aspects, and activities related to daily living. Possible responses on each item are 0 or 1; higher total scores (0–25) indicate more problems with daily functioning. The total score was used to determine frailty, with scores of ≥8 indicating frailty, 4–7 pre-frailty, and ≤3 robustness [26]. As frailty includes physical, mental, and social factors, the KCL was selected because it is a multifactorial index. Participants with KCL total scores of 4 to 7 were categorized as the pre-frailty group, and those with scores of ≤3 as the healthy group.

#### 2.2.2. Social Aspects

Household composition, employment status, economic status, frequency of going out, presence of intimate friends, community participation in middle age, social activities, solitary eating, and presence of friends who listen to one’s concerns were assessed.

#### 2.2.3. Physical Aspects

The history of disease, height, weight, regular exercise habits, and food intake status were ascertained. If a participant has been diagnosed with any chronic or serious disease in the past, i.e., hypertension, diabetes, respiratory disease, cardiovascular disease, stroke, cancer, depression, or dementia, the history of disease was “yes”. Body mass index (BMI kg/m^2^) was calculated from height and weight, and a BMI of <18.5 was defined as “thin”. Food intake status was determined using the Dietary Variety Score (DVS) [27] for 10 foods or food groups (seafood, meat, eggs, milk and dairy products, legumes, green and yellow vegetables, seaweed, potatoes, fruit, and oils and fats). Responses were on a 4-point scale ranging from “eat every day” to “rarely eat”, with 1 point given for “eat every day” and 0 for all other responses. The total score ranged from 0 to 10.

#### 2.2.4. Mental Aspects

Poor self-rated health and having depressive status [28] were assessed. To measure self-rated health views, the respondents were asked to choose only one option from the following: (1) excellent, (2) good, (3) not good, (4) poor. The response patterns were further grouped into a binary, with “excellent”, and “good” as one group and the rest as “otherwise”. The presence of depressive status was defined as a “yes” response to either the question “I feel down or depressed” or the question “I am just not interested in things” during the last month.

#### 2.2.5. Basic Attributes

The basic attributes of age and sex (as being biologically defined) were determined.

### 2.3. Statistical Analyses

A Multivariable logistic model was used to explore demographic, physical, and mental factors associated with the presence or absence of pre-frailty as the dependent variable. Among the social factors, solitary living, having a job, economic status, having intimate friends, being invited by someone, sources of health information, outings at least once a week, and listening to a friend’s concerns were excluded, and community participation in middle age, social participation at least once a week, listening to one’s concerns, and solitary eating were included because the status of these factors can be changed by intervention. Based on the perspective that there were gender differences in the association of some social factors, we determined to adopt models including interaction terms with sex, while both sexes were included as samples to increase the power to detect risk factors common to both. We also examined interaction terms with sex for all confounders and candidate social factors, and we excluded from the model main effects and interaction terms with weak associations (*p* > 0.20). Model building was conducted in an exploratory manner based on the AIC, oriented toward a model with the minimum number of parameters (a “parsimonious” model), and the final model was determined.

In the final model, odds ratios (ORs) and 95% confidence intervals (CIs) were estimated for factors associated with pre-frailty. For some factors, the same ORs were estimated for both genders, and for some factors with the interaction terms that were statistically significant, the ORs of association with pre-frailty were estimated for each gender. For model validation, the Hosmer–Lemeshow test was conducted to evaluate goodness of fit, and the c statistic was calculated to evaluate a predictive performance.

Continuous variables were reported as the mean ± standard deviation (SD), and categorical variables were reported as numbers and percentages. Differences in social, physical, and mental dimensions, and basic attributes, in the two groups by gender were confirmed using the chi-square test or Fisher’s exact test (for nominal variables) and the *t*-test (for continuous variables). The two-tailed tests were used to determine the significance at the 5% level. The statistical package SAS version 9.4 (SAS Institute Inc., Cary, NC, USA) was used for statistical analysis.

### 2.4. Ethical Considerations

The research was explained to the participants and their written informed consent was obtained. The study was approved by the ethics review committee of Teikyo University School of Medicine (Teirin 21-129). The study was carried out in accordance with relevant guidelines and regulations, including the Declaration of Helsinki and the ethical guidelines for epidemiological research, published in collaboration with the Ministry of Education, Culture, Sports, Science and Technology and the Ministry of Health, Labour and Welfare, Japan.

## 3. Results

Of the 494 participants who returned questionnaires (54.9% response rate), 384 (203 men and 181 women) were included in the analysis. The analysis excluded 1 case with missing data for gender and age and 109 cases (32 men and 77 women) that were categorized as frail.

### 3.1. Comparison of Pre-Frailty and Healthy Groups (Table 1)

#### 3.1.1. Participant Characteristics

Based on the KCL scores, the percentage of women respondents classified as pre-frailty was 103/181 (57%), which was higher than the 93/203 (46%) among men.

The mean (standard deviation) age of the pre-frailty group was 75 (6) years for men and 75 (6) years for women.

**Table 1 ijerph-20-01091-t001:** Comparison of Japanese community-dwelling older adults in the pre-frailty and healthy groups.

			Men							Women						
			Total		Pre-Frailty		Healthy		*p*	Total		Pre-Frailty		Healthy		*p*
			(n = 203)		(n = 93)		(n = 110)			(n = 181)		(n = 103)		(n = 78)		
			n (%) or Mean ± SD		n (%) or Mean ± SD		n (%) or Mean ± SD			n (%) or Mean ± SD		n (%) or Mean ± SD		n (%) or Mean ± SD		
Demographic	Age (mean ± SD)		74	±6	75	±6	74	±5	0.089	74	±6	75	±6	73	±5	0.019
Physical	DVS (mean ± SD)		2.9	±2.4	2.6	±2.3	3.3	±2.4	0.075	3.9	±2.5	3.9	±2.7	3.8	±2.3	0.776
	Exercise habit	yes	153	(77)	65	(71)	88	(81)	0.094	146	(84)	81	(82)	65	(87)	0.389
		no	46	(23)	26	(29)	20	(19)		28	(16)	18	(18)	10	(13)	
	History of disease	yes	40	(22)	19	(23)	21	(22)	0.933	37	(22)	16	(17)	21	(30)	0.042
		no	143	(78)	69	(77)	74	(78)		129	(78)	80	(83)	49	(70)	
	BMI	<18.5	10	(5)	8	(9)	2	(2)	0.047	17	(10)	10	(10)	7	(10)	0.941
		≥18.5	187	(95)	84	(91)	103	(98)		152	(90)	88	(90)	64	(90)	
	Subjective symptoms	no	109	(57)	44	(48)	65	(66)	0.010	106	(62)	51	(37)	55	(74)	0.003
		yes	81	(43)	48	(52)	33	(34)		66	(38)	47	(63)	19	(26)	
Mental	Depressive state	yes	31	(16)	22	(24)	9	(8.9)	0.004	46	(27)	34	(35)	12	(16)	0.004
		no	160	(84)	68	(76)	92	(91)		126	(73)	62	(65)	64	(84)	
	self-rated health	better	177	(93)	78	(87)	99	(98)	0.003	156	(89)	84	(85)	72	(95)	0.037
		poor	14	(7)	12	(13)	2	(1.9)		19	(11)	15	(15)	4	(5)	
Social	Solitary living	yes	23	(11)	12	(13)	11	(10)	0.629	25	(14)	17	(17)	8	(10)	0.208
		no	178	(89)	79	(87)	99	(90)		154	(86)	84	(83)	70	(90)	
	Solitary eating	no	162	(81)	71	(76)	91	(85)	0.118	139	(77)	73	(72)	66	(85)	0.039
		yes	38	(19)	22	(24)	16	(15)		41	(23)	29	(28)	12	(15)	
	Job	yes	91	(45)	38	(41)	53	(48)	0.328	43	(24)	22	(22)	21	(28)	0.370
		no	111	(55)	54	(59)	57	(52)		134	(76)	79	(78)	55	(72)	
	Economic status	difficult	55	(27)	31	(34)	24	(22)	0.113	39	(22)	23	(23)	16	(21)	0.728
		usually	125	(62)	53	(58)	72	(65)		130	(73)	74	(73)	56	(73)	
		afford	21	(11)	7	(8)	14	(13)		9	(5)	4	(3.9)	5	(6)	
	Outings at least once a week	yes	189	(95)	88	(97)	101	(94)	0.511	173	(97)	97	(96)	76	(99)	0.391
		no	9	(5)	3	(3)	6	(6)		5	(3)	4	(3.9)	1	(1)	
	Intimate friend	yes	165	(85)	74	(82)	91	(87)	0.391	158	(91)	88	(89)	70	(93)	0.315
		no	30	(15)	16	(18)	14	(13)		16	(9)	11	(11)	5	(7)	
	Being invited by someone	yes	91	(47)	42	(48)	49	(46)	0.835	106	(60)	55	(54)	51	(66)	0.113
		no	103	(53)	46	(52)	57	(54)		72	(40)	46	(46)	26	(34)	
	Sources of health information	family and friend	106	(54)	46	(51)	60	(57)	0.442	118	(66)	62	(61)	56	(73)	0.113
		otherwise	90	(46)	44	(49)	46	(43)		60	(34)	39	(39)	21	(27)	
	Community participation in middle age	yes	117	(60)	49	(55)	68	(64)	0.197	102	(59)	49	(50)	53	(71)	0.006
		no	78	(40)	40	(45)	38	(36)		71	(41)	49	(50)	22	(29)	
	Participation in social activities at least once a week	yes	96	(49)	41	(45)	55	(53)	0.245	79	(47)	40	(43)	39	(52)	0.221
		no	100	(51)	51	(55)	49	(47)		90	(53)	54	(57)	36	(48)	
	Friends who listen to one’s concerns	yes	60	(30)	30	(33)	30	(28)	0.403	102	(58)	53	(52)	49	(64)	0.110
		no	140	(70)	61	(67)	79	(72)		75	(42)	48	(48)	27	(36)	
	Listening to a friend’s concerns	yes	64	(32)	31	(34)	33	(31)	0.560	102	(59)	54	(55)	48	(64)	0.210
		no	134	(68)	59	(66)	75	(69)		72	(41)	45	(45)	27	(36)	

DVS, Dietary Variety Score; BMI, body mass index. History of disease: If a participant had been diagnosed with any chronic or serious disease in the past, i.e., hypertension, diabetes, respiratory disease, cardiovascular disease, stroke, cancer, depression, or dementia, the history of disease was “yes”.

#### 3.1.2. Social Aspects

The women in the healthy group were more likely to participate in community activities in middle age (*p* = 0.006). In contrast, no differences were found for men.

#### 3.1.3. Physical Aspects

Among men, the proportion of those with a BMI less than 18.5 (so-called “thin”) was higher in the pre-frailty group than in the healthy group (*p* = 0.047), and the proportion of those who reported no subjective symptoms was higher in the healthy group (*p* = 0.015). For the women, the percentage of those who reported no subjective symptoms was also higher in the healthy group (*p* = 0.005).

#### 3.1.4. Mental Aspects

For both men and women, a lower percentage of those in the healthy group were classified as depressed (*p* = 0.004). In addition, a higher percentage of men in the healthy group reported better self-rated health (*p* = 0.003).

### 3.2. Factors Associated with Pre-Frailty

The factors positively and negatively associated with pre-frailty considered common to both genders are presented in Table 2, as well as the positive and negative associated factors specific to men and the negative associated factors specific to women. The factors associated with pre-frailty for both men and women and their ORs (95% CIs) were older age (≥75 years old): 1.96 (1.23–3.12), depressive status: 3.39 (1.84–6.25), no community participation in middle age: 1.84 (1.16–2.92), poor self-rated health: 3.60 (1.32–9.78), and having subjective symptoms: 2.05 (1.27–3.29).

The gender-differential characteristics of factors associated with pre-frailty were BMI (less than 18.5): 7.69 (1.50–39.56) and DVS (three or more food groups consumed daily): 0.66 (0.44–0.98) for men. For women, having friends who listen to one’s concerns was 2.54 (1.26–5.10) (Table 2 and Figure 1).

The final model (c statistic = 0.749) showed a good fit with a chi-square value (degrees of freedom) of 7.7 (8), *p* = 0.463 in the Hosmer–Lemeshow test.

## 4. Discussion

This study focused on socio-demographic factors and aimed to examine potential influencing factors for pre-frailty. The results of this study were as follows. For both genders, no community participation in middle age, identified as a socially relevant factor, was associated with health (and negatively associated with pre-frailty). For women, having friends who listen to one’s concerns was found to be positively associated with health (but negatively associated with pre-frailty). In other words, those who did not participate in the community in middle age for both genders, and those who did not have friends who listened to their concerns in the case of women, were more likely to be classified as pre-frailty. These findings persisted even after adjusting for other factors, such as those listed in Table 2. However, we did not identify any pre-frailty-associated social factors for men alone. The effect of gender was consistent across candidate models and for factors such as age, depressive state, exercise habit, and self-rated health, interaction terms with sex were not statistically significant, and there were no gender differences in the strength of the frailty association. However, the results were dependent on the sample size of the present study, and the possibility remains that gender differences will be found for the factors for which no gender differences found in this study as a result of data analysis with a larger sample size.

Physical and psychological factors associated with pre-frailty included age for both genders. Having subjective symptoms, depressive status, and poor self-rated health, along with a low DVS score, were also factors for men. Although not shown in Table 2, the OR (95% CI) for pre-frailty status for women relative to men was 0.56 (0.21–1.51), with no statistically significant gender difference. However, the causal relationship between factors that may influence pre-frailty and its manifestation was not clear, so results should be interpreted carefully.

The association between participation in the community in middle age and frailty has not been reported to date. The following are possible mechanisms by which participation in the community in middle age may prevent the onset of frailty. Older frail people are known to have fewer opportunities to go out than healthy older people [29], while there is evidence that those who participate in community activities in middle age are more likely to continue to participate in community activities after retirement, because they have more resources that are likely to lead to continued community activities, such as having a familiar place to engage in activities and having more experience interacting with friends and acquaintances during community activities [30]. Therefore, it is important to provide opportunities for both men and women to get involved with residents before they reach older age, so that they can continue to be involved with the community during older age. For example, a feasible intervention policy would be to revitalize existing community facilities for social activities and to involve middle- to older-aged individuals in such activities [31]. It is also important to conduct various awareness-raising activities regarding the importance of the local and regional community, not only for the elderly, but for the entire local population.

Having friends who listen to one’s concerns is an emotional aspect of social support. Social support affects health and has been confirmed as having a positive effect on the health of older people, including an inhibiting effect on early death [32,33]. Compared with men, women tend to have more close friends and to consult friends about their concerns [33,34]. In addition, men’s social networks tend to be biased toward their spouses, whereas women’s social networks are more diverse and include children, other relatives, and friends [32]. In this study, we confirmed that men tend to have less emotional support than women. However, the effects of social and emotional support on frailty are unclear [10]. There is evidence for an association between emotional support and depression [35] and for an association between depression and frailty [36]. This suggests that social and emotional support may be associated with frailty, although it remains to be confirmed whether emotional support has a direct effect on pre-frailty.

However, research shows that many people experience a lack of companionship and trusted friends as they get older, and that elderly people living alone particularly lack emotional support [37]. A lack of social support is associated with depression [37]; therefore, individuals need a variety of ways in which they can acquire emotional support.

The results support previous findings showing an association between aging [38], having depression [36], poor self-rated health [39,40], having subjective symptoms [41], and frailty. The present results were consistent with previous findings for men showing that thinness [42] and DVS scores [27,43] are associated with frailty. The lack of these associations for women may be because health awareness and health consciousness tend to increase with age, particularly in women [44,45]. Additionally, even women with pre-frailty tend to practice healthy behaviors, such as good eating habits and maintenance of an appropriate weight in accordance with such habits. This may explain the gender difference in the present study.

Other social factors, including solitary eating, going out more than once a week [46], and participating in social activities, have been previously reported to affect frailty; however, these were not identified as relevant factors in the present study. This may have been because few of our participants reported solitary eating or not going out more than once a week. In addition, participation in social activities can be affected by area of residence [11]; the geography of Nagatoro, which is surrounded by mountains, may therefore have affected participants’ social activities. Furthermore, there are many different types of social activities that have differing effects on health [47], so the type of activity engaged in needs to be considered. There may be other important social factors that were not investigated in this study. In addition, the limited number of participants and skewed distribution of responses may have contributed to the lack of association of known influencing factors in this study. Therefore, further research is needed to explore the effect of additional factors.

This study had several limitations. First, because this was a cross-sectional study, it was not possible to determine causal relationships between factors thought to affect pre-frailty and its development. Longitudinal observation studies or intervention studies should be conducted to confirm causality. In particular, the possibility of reverse causality, in which unhealthy individuals are less likely to have social connections, has been reported [10], and further research on social factors is needed.

Second, we were not able to confirm whether the experience of community participation in participants’ middle age was linked to continued community participation in older age. Therefore, additional studies are needed to investigate the continuity of community participation.

Third, generalizability is limited because this study was conducted with older adults in one specific region, and regional characteristics may have affected the results. Furthermore, the response rate for the questionnaires used in this study was not very high. Coupled with the small sample size, generalizability is limited. However, this is the first study to examine the possibility of preventive intervention from a social aspect, taking gender into account, and the findings may be useful for future studies of interventions to prevent or improve frailty.

Finally, the study has been developed in a historical moment influenced by the COVID-19 pandemic. In a private correspondence from a public health nurse in the community where the survey was conducted, we were told that during the COVID-19 outbreak, residents, including the elderly, tended to refrain from going out. Therefore, we cannot omit the impact of COVID-19 on frailty.

## 5. Conclusions

This cross-sectional study of older adults showed that no community participation in middle age is a social factor that is associated with pre-frailty. From the viewpoint of gender, having a friend to listen to concerns is associated with pre-frailty for women. The findings suggest that in order to prevent frailty and pre-frailty, it is necessary to provide support that emphasizes social aspects, such as having developed community connections before older age, and personal relationships that provide emotional support. However, further research on social factors among men is warranted.

## Figures and Tables

**Figure 1 ijerph-20-01091-f001:**
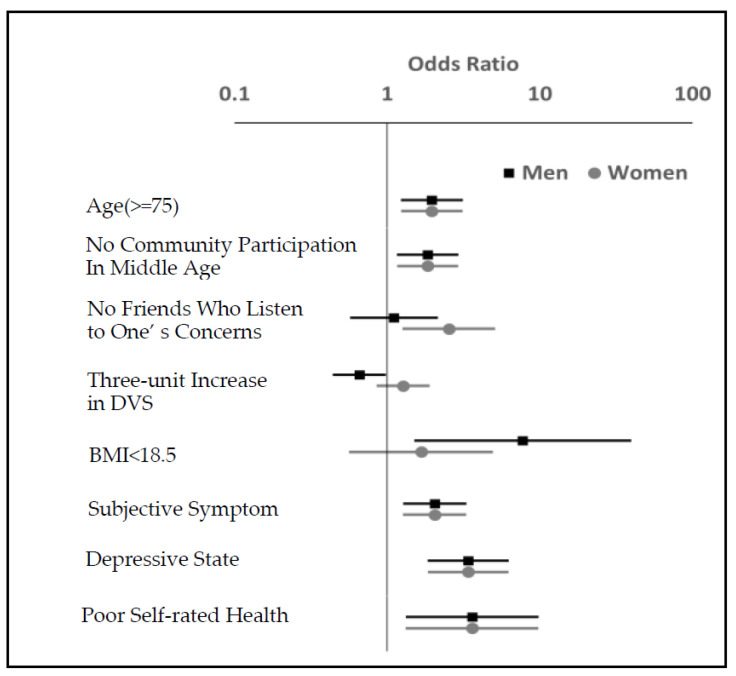
Gender differences associated with pre-frailty. The estimated odds ratios in Table 2 are visually displayed.

**Table 2 ijerph-20-01091-t002:** Factors associated with pre-frailty in Japanese rural community-dwelling older adults.

		Men (n = 203)		Women (n = 181)	
		OR	(95% CI)	OR	(95% CI)
Demographic	Age				
	≥75	1.96	(1.23–3.12)	Same as OR in men	
	65–74	1			
Social	Community participation in middle age				
	no	1.84	(1.16–2.92)	Same as OR in men	
	yes	1			
	Friends who listen to one’s concerns				
	no	1.05	(0.54–2.05)	2.54	(1.26–5.10)
	yes	1		1	
Physical	DVS				
	Three more groups of food	0.66	(0.44–0.98)	1.27	(0.85–1.90)
	BMI				
	<18.5	7.69	(1.50–39.56)	1.67	(0.56–4.93)
	≥18.5	1		1	
	Subjective symptoms				
	yes	2.05	(1.27–3.29)	Same as OR in men	
	no	1			
Mental	Depressive state				
	yes	3.39	(1.84–6.25)	Same as OR in men	
	no	1			
	Self-rated health				
	poor	3.60	(1.32–9.78)	Same as OR in men	
	better	1			

DVS, Dietary Variety Score; BMI, body mass index; OR, odds ratio; CI, confidence interval. The odds ratios were calculated using a model that included all explanatory variables.

## Data Availability

The datasets generated and analyzed during the current study are available from the corresponding author on reasonable request.

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
