# Peer review of "Gender Differences in Socio-Demographic Factors Associated with Pre-Frailty in Japanese Rural Community-Dwelling Older Adults: A Cross-Sectional Study"

_ijerph, 2023, doi:10.3390/ijerph20021091_

Round 1

Reviewer 1 Report

Dear authors, 

thank you for the opportunity to read your paper. I concur with the overall conclusions of your papers which are in line with research with the same topic in my country. However, I recommend a thorough read-through so that the correct conclusions appear consistently throughout the paper. In my reading I think that a 'no' in combination with 'community participation' often is missing, which initially makes the reader think that community participation is something that is negative. I would recommend some type of abbreviation here so that it is clear in the paper that you either refer to CP community participation or nCP no community participation. Good luck with the final revisions. 

Reviewer 2 Report

Congratulations for your work, I have some constructive comments to consider: 

- Lines 34 - 47: Between these lines, when reading, maybe is interesting to consider to use some synonyms of these two terms to avoid repeating them too much in the same paragraphs. 

Frairly:  appears 14 times. 

Pre-frailty: appears 8 times. 

Reviewer 3 Report

The research topic of this paper is of value to countries and regions with serious aging problem, and is a very interesting problem. The paper is clear in logic and structure and rigorous in research methods and empirical analysis. I have some comments that I would like the authors to address:

(1) Please explain the reasons for selecting Nagatoro town as the research object in the Materials and Methods section and illustrate its generality. The existence of the aging problem doesn’t make Nagatoro town representative of the problem. I recommend that the authors use a map to mark the location of Nagatoro town in Japan so that international readers can better understand the situation.

(2) I recommend that the authors delete the sentence of “Nagatoro is located in a basin surrounded by mountains” in Line 71 from the manuscript. The above sentence is irrelevant to the research.

(3) The paper mainly uses forest plot to exhibit the confidence interval of the coefficients. I recommend the authors to provide more detailed explanations on the empirical results in the main body or appendix of the paper, especially explanations on the selection of model and its final form. In addition, I am wondering if the authors have considered the issue of heteroscedasticity for the cross-section data, which is whether gender will lead to increases in variance. Please give more detailed explanations.

(4) The Conclusions section should better summarize the empirical results and discussion, and illustrate the significance and implications of the research.

Author Response

We would like to express our deep appreciation for the comments and suggestions. If you require further revisions that would make this paper acceptable, we would be glad to make them. We certainly appreciated your very careful review and would be more than happy to follow any further suggestions. Thank you again for considering our manuscript.

Point 1: Please explain the reasons for selecting Nagatoro town as the research object in the Materials and Methods section and illustrate its generality. The existence of the aging problem doesn’t make Nagatoro town representative of the problem. I recommend that the authors use a map to mark the location of Nagatoro town in Japan so that international readers can better understand the situation.

Response 1: Thank you for your comment on this point. We added the following sentence in the 1st paragraph in Materials and methods:

“ In Nagatoro-town, where the population is expected to age further in the future, measures to prevent frailty are an urgent issue. In a depopulated area with an aging population such as Nagatoro-town, it is expected that many of the residents' life back-grounds are similar, and the results of this study may be adapted if a certain level of evaluation is obtained in similar areas.”

Point 2:  I recommend that the authors delete the sentence of “Nagatoro is located in a basin surrounded by mountains” in Line 71 from the manuscript. The above sentence is irrelevant to the research.

Response 2: Thank you very much for your thoughtful suggestion. We deleted those sentences, pointed out by the reviewer.

Point 3: The paper mainly uses forest plot to exhibit the confidence interval of the coefficients. I recommend the authors to provide more detailed explanations on the empirical results in the main body or appendix of the paper, especially explanations on the selection of model and its final form. In addition, I am wondering if the authors have considered the issue of heteroscedasticity for the cross-section data, which is whether gender will lead to increases in variance. Please give more detailed explanations.

Response 3: Thank you very much for your thoughtful suggestion.  Figure 1 and Table 2 show the same thing, but we apologize for the lack of explanation. The forest plot in Figure 1 visually displays the estimated values of the model parameters presented in Table 2. To make it easier for readers, we added references to Figure 1, where Table 2 was referred to in the text, and modified a caption for Figure 1. Also, we have aligned the order of the factors in Figure 1 to the same as Table 2.

caption of Figure 1

Before modification: Figure 1. Gender differences associated with prefrailty.

After modification: Figure 1. Gender differences associated with prefrailty. The estimated odds ratios in Table 2 were visually displayed.

Point 4:  The Conclusions section should better summarize the empirical results and discussion, and illustrate the significance and implications of the research.

Response 4: Thank you very much for your informative suggestion. According to the suggestion, we have corrected it.

The modified sentence reads as follows:

After modification:

“This cross-sectional study of older adults showed that no-community participation  in middle age is a social factor that affects prefrailty. From the viewpoint of gender, having a friend to listen to concerns affected prefrailty for women. The findings suggest that in order to prevent frailty and prefrailty, it is necessary to provide support that emphasizes social aspects, such as having develop community connections before older age, and personal relationships that provide emotional support. However, further research on social factors among men is warranted.”

Before modification:

“This cross-sectional study of older adults showed that community participation in middle age is a social factor that affects prefrailty. From the viewpoint of gender, having a friend to listen to concerns affected prefrailty for women. The findings suggest the need for support, with an emphasis on social aspects, to prevent frailty and prefrailty. In particular, it is important for individuals to have a variety of personal relationships that provide emotional support, and to develop community connections before older age. However, further research on social factors among men is warranted.”

Regarding your doubts about heteroscedasticity, the logistic model does not require the assumption of homoscedasticity, i.e., that the residuals follow a normal distribution with equal variance regardless of the set of covariates. Since we did not know what your suspicions were, we provide a table of model parameter estimates for several candidate models, including the final model, for your reference. The effect of gender was consistent across candidate models and for factors such as Age, Depressive state, Exercise habit, and Self-rated health, interaction terms with sex were not statistically significant, and there were no gender differences in the strength of the frailty association. However, the results were dependent on the sample size of the present study, and the possibility remains that gender differences will be found for the factors for which no gender differences found in this study as a result of data analysis with a larger sample size. The limitation on this point has been described in line 293 and added this time in line 215 in file " ijerph-2073335_Tracked.

Reviewer 4 Report

I had an opportunity to review the manuscript entitled “Gender Differences in Socio-Demographic Factors Associated 2 with Prefrailty in Japanese Rural Community-Dwelling Older 3 Adults: A Cross-Sectional Study”.

Please find my comments and suggestions below:

1.       Page 2, line 68, stratified by?

2.       A multivariate logistic model was used to explore demographic, physical, and mental factors associated with the presence or absence of prefrailty as the dependent variable”.

Multivariable models were performed.

3.       Please explain, why the “frail” group was not analyzed? (more than 20% of the whole sample)

4.       Table 1, for numerical variables, please use “±” not “/”,

5.       Please explain why the interaction between factors and gender were analysed.

It’s not clear why, e. g.  in table 2, for age category, one OR is presented, but in Figure 1 two ORs were presented. It might be worth providing a P value for interaction on Figure 1.

6.       Page 4, Lines 180-181, in statistical analysis, please add information that these statics were used to described the goodness of fit. Moreover, I suggest using C-statics instead of e. g. the Hosmer–Lemeshow test.
